# Fucoidan Characterization: Determination of Purity and Physicochemical and Chemical Properties

**DOI:** 10.3390/md18110571

**Published:** 2020-11-19

**Authors:** Ahmed Zayed, Mona El-Aasr, Abdel-Rahim S. Ibrahim, Roland Ulber

**Affiliations:** 1Institute of Bioprocess Engineering, Technical University of Kaiserslautern, Gottlieb-Daimler-Straße 49, 67663 Kaiserslautern, Germany; ahmed.zayed1@pharm.tanta.edu.eg; 2Department of Pharmacognosy, Tanta University, College of Pharmacy, El-Guish Street, Tanta 31527, Egypt; moelaasar@pharm.tanta.edu.eg (M.E.-A.); abdelrahim.ibrahim@pharm.tanta.edu.eg (A.-R.S.I.)

**Keywords:** fucoidans, fucoidanases, glycosidic linkages, molecular masses, NMR, structure–activity relationships

## Abstract

Fucoidans are marine sulfated biopolysaccharides that have heterogenous and complicated chemical structures. Various sugar monomers, glycosidic linkages, molecular masses, branching sites, and sulfate ester pattern and content are involved within their backbones. Additionally, sources, downstream processes, and geographical and seasonal factors show potential effects on fucoidan structural characteristics. These characteristics are documented to be highly related to fucoidan potential activities. Therefore, numerous chemical qualitative and quantitative determinations and structural elucidation methods are conducted to characterize fucoidans regarding their physicochemical and chemical features. Characterization of fucoidan polymers is considered a bottleneck for further biological and industrial applications. Consequently, the obtained results may be related to different activities, which could be improved afterward by further functional modifications. The current article highlights the different spectrometric and nonspectrometric methods applied for the characterization of native fucoidans, including degree of purity, sugar monomeric composition, sulfation pattern and content, molecular mass, and glycosidic linkages.

## 1. Introduction

Marine polysaccharides are classified as sulfated (SPs) and nonsulfated macromolecules that are mainly derived from micro- and macroalgae [1,2]. In particular, macroalgal SPs show more diverse chemical characteristics than nonsulfated analogues, in terms of their molecular weight, monosaccharide composition, and sulfate content and position, which interact with various biological targets at different levels leading to diverse and promising pharmacological activities [2,3]. They are found in different phyla, such as in Phaeophytes or brown algae (e.g., fucoidans), Rhodophytes or red algae (e.g., carrageenan), and Chlorophytes or green algae (e.g., ulvan) [2,4]. Despite brown algae, with 1755 species, not being the most abundant class, their SP fucoidans show more potential applications in different areas than those isolated from ulvan and carrageenan [5,6,7].

Fucoidans are known as fucose-containing sulfated polysaccharides (FCSPs), where l-fucose always predominates other sugar monomers, such as galactose, mannose, glucose, and uronic acids. l-fucose may exceed 90% of the total sugar composition of fucoidans [8]. Yet, galactose, as in the case of sulfated galactofucans, may possess similar ratios to fucose [9]. Another type of FCSP is isolated from marine invertebrates, called sulfated fucans. In contrast, they are composed of l-fucose only. Hence, the term fucoidans has recently been adopted specifically for the heterogenous marine SPs rich in fucose and derived from the different species of brown algae, including the old names fucoidin and fucoidan, to be consistent with the International Union of Pure and Applied Chemistry (IUPAC) nomenclature system [10]. They are species-specific and, therefore, they do not have a universal chemical structure. Yet, they represent the major component of cell walls and the extracellular matrix (ECM) along with alginate and cellulose in brown seaweeds [11,12]. The physicochemical and chemical heterogeneity of fucoidans was discussed previously, as well as the way in which it affects their application [13]. In addition, fucoidans are characterized by high molecular weights, up to 950 kDa in the native fucoidan of *Hizikia fusiforme* or *Sargassum fusiforme* [14]. The presence of sulfate ester groups imparts a negative charge on the macromolecule skeleton responsible for the anionic characteristic of fucoidans [15]. Moreover, chain branching increases the complexity of fucoidans compared to sulfated fucans derived from marine invertebrates [16,17]. Therefore, an investigation of these two groups of fucose-containing biopolymers, i.e., sulfated fucans and fucoidans, requires different investigational approaches. Owing to their complicated chemical structures, enzymatic hydrolysis, mild acid hydrolysis, and autohydrolysis of native fucoidans are always involved in the elucidation of their fine structural features. Such pretreatments enable the production of oligomers or simple fractions that are easily interpreted [18].

Various applications of fucoidans in the therapeutic [19], cosmeceutical [20], nutraceutical/functional foods [21], diagnostic [22], and drug delivery [23] fields have increased awareness concerning their importance, especially in the last few decades. Specifically, the pharmacological activities of fucoidans make them a candidate for the treatment of bleeding disorders [24], inflammation [25], viral infections [26], malignant tumors [27], and immune disorders [28]. Hence, current studies are focusing on pharmacokinetic and tissue distribution investigations after oral and topical administration of fucoidans [29,30,31]. These studies may help the administration of fucoidans in prospective human clinical trials. Meanwhile, the modification of native fucoidans via chemical or enzymatic treatment may result in an increase in their biological activity [11,32].

The mechanisms involved in the biological interactions of fucoidans with various targets are not only based on charge density, but also fine chemical features [33]. Hence, the major obstacle for the determination of structure–activity relationships is the complex structures of such polysaccharides [34]. These activities are highly correlated with the structural and nonstructural features of fucoidans [35,36], as summarized in Figure 1. Examples include relationships between high molecular weight and anticoagulant activity [37,38], low molecular weight and cancer apoptosis [39], and selective angiogenic activity in tumors as antiangiogenic [40] or in impaired tissues as proangiogenic agents [41]. Moreover, it was reported that the sulfation pattern was an important factor, especially at C-4 for anti-Herpes simplex virus infection [42] or at C-2 and C-3 for anti-coagulant activity [43], as well as the sulfate content (charge density) for cytotoxic and anticancer activity [44,45,46,47] or repair potential of injured kidney cells [48], the branching degree for antitumor activity [49], and the monosaccharide composition in terms of uronic acid content for antioxidant activity [50]. Moreover, more than one factor may be involved, such as the sulfate/fucose molar ratio, as shown in the attenuation of oxidative stress-induced cellular cytotoxicity by the crude fucoidan prepared from *Sargassum crassifolium* [51], or the high sulfate and low uronic acid content for significant anticoagulant activity as shown by *Ecklonia cava* fucoidan [11,52].

In addition, the quality and content of fucoidan in commercial products can affect its use. The most well-known example constitutes its cosmeceutical applications, which are improved by the presence of phlorotannins owing to the antioxidant activity of polyphenolics [53]. Hence, the quantitative determination of fucoidans is highly important for the prevention of interferences from coextracted contaminants when investigating the bioactivities and assuring the quality of commercially available products [54,55].

Therefore, the qualitative and quantitative determination of fucoidans and their coextracted contaminants is a must before physicochemical, chemical, and biological characterization. Afterward, the other functional motifs of fucoidan should be well characterized, such as its molecular weight, monomeric composition, sulfate content and pattern, and glycosidic linkage. Lastly, relationships with certain activities may be identified, and the putative mechanisms of action can be constructed to explain such activities. The current article focuses on the possible tests and investigations which help in the full characterization of fucoidans after downstream processing.

## 2. Chemistry of Fucoidans

The chemistry of fucoidans is highly variable according to their origin, especially regarding their complexity. For instance, fucoidans derived from seaweeds commonly show a branched and more sulfated skeleton with the presence of numerous sugar monomers in addition to *α*-l-fucose. However, marine invertebrate fucoidans, such as echinoderms (e.g., sea cucumber) and urchins (e.g., *Strongylocentrotus droebachiensis*), are less complex, consisting of a linear and regular chain of repeating *α*-l-fucose units [10,16,56,57,58]. These differences make algal fucoidans a more preferable biogenic resource than those from marine invertebrates, in addition to their multiple and interesting biological activities [59].

In the literature, several structural models for seaweed fucoidans have been suggested to describe their important structural features [6,60,61,62] depending on their macroalgal biogenic origin, including species, age, geographical origin, and season of harvesting [9,13,63]. Cao et al. presented several representative fucoidan structures isolated from *Fucus evanescens*, *Fucus vesiculosus*, *Sargassum mcclurei*, *Turbinaria ornata*, *Saccharina cichorioides*, and *Undaria pinnatifida* [9].

Nevertheless, the most widely accepted models are those introduced by Cumashi et al. [64] and Ale et al. [35]. They proposed that seaweed fucoidans are highly heterogenous within brown seaweed species, composed of a linear or branched sulfated l-fucopyranoside backbone linked by not only alternating *α*-(1→3) and *α*-(1→4) linkages, but also *α*-(1→4) and *α*-(1→3) linkages only. Other sugar monomers can also be found, such as *β*-d-galactose, *β*-d-mannose, *α*-d-glucuronic acid, *α*-d-glucose, and *β*-d-xylose, but their positions and binding modes are still not understood [64,65]. However, Bilan et al. studied a fucoidan fraction isolated from *Sargassum polycystum* (Fucales) and found 2-linked sulfated *α*-d-galactopyranose residues [62]. Moreover, the l-fucose unit is mono- or disulfated and may be acetylated. These groups are responsible for the anionic characteristic of fucoidans.

According to the models proposed by Cumashi et al. and Ale et al., the chemical structures of fucoidans are represented on the basis of their origin, as shown in Figure 2. For examples, in Fucales, fucoidans show l-fucopyranoside chains linked with alternating *α*-(1→4) and *α*-(1→3) glycosidic linkages. C-2 and/or C-4 (rarely at C-3) are usually substituted with sulfate ester groups (–SO_3_^−^), according to the type of glycosidic linkages [9]. Moreover, side branching was detected at C-4, alternating with sulfate groups in *F. serratus* L. in *α*-(1→3) l-fucopyranoside units. On the other hand, in Laminariales and Chordariales, fucoidan subunits are mainly linked by *α*-(1→3) glycosidic linkages. Additionally, at C-2, other sugar monomers can be detected as a side branch, whereas sulfate ester groups are common at C-4. The chemical structures may also vary within the same organism on the basis of the applied extraction methods [66].

Recently, Usoltseva et al. revealed other models in Laminariales members, i.e., *Saccharina* or *Laminaria cichorioides* and *Laminaria longipes*. They detected unusual fucoidans with *α*-(1→3) linkages that also contain *α*-(1→4)- and *α*-(1→2)-linked fucopyranoside residues [67]. Additionally, Wang et al. showed that *α*-(1→4) linkages may be present in the fucoidan backbone of *Laminaria japonica* [68]. As a consequence of these complex characteristics and heterogeneity, it is always difficult to characterize the chemical structure of the whole polymer using a single technique. Spectrometric methods (e.g., Fourier-transform infrared (FT-IR), NMR and MS) are used to elucidate the structural features, especially the position of sulfate ester groups and glycosidic bonds. In addition, chromatographic methods, such as gel permeation (GPC) also known as size-exclusion chromatography (SEC), are applied for the determination of molecular-weight parameters or averages. Currently, advanced hyphenated spectrometric techniques, such as HPLC–MS/MS, are applied [69,70]. Furthermore, the application of regio- and stereoselective enzymatic degrading fucoidans isolated from marine bacteria provided new insight into the chemical structure of fucoidan, when combined with spectrometric methods [6,14]. These items are discussed in Section 6.

## 3. Characterization of Fucoidan Quality

### 3.1. Fucoidan Characteristics

#### 3.1.1. Sugar Content

The Dubois or phenol–sulfuric acid assay is a simple acid-catalyzed condensation reaction, which is commonly employed for the determination of total sugar concentration in carbohydrates [71]. Fucoidan and 5% (*w/v*) aqueous phenol solutions are mixed, and then concentrated sulfuric acid is carefully added. Afterward, the sample is mixed vigorously, and the absorbance is recorded at 490 nm. The reaction mechanism is based on color development upon the dehydration of sugars to furfural derivatives with sulfuric acid. The furfural product is then condensed with phenol to produce stable colored compounds.

The Somogyi–Nelson test is also used for the determination of reducing sugars, where copper and fucoidan solutions are mixed carefully and incubated in a boiling water bath. Afterward, the arsenic molybdate reagent is added. The reaction mixture is then incubated at room temperature and analyzed at 500 nm. The mechanism is based on a redox reaction, where reducing sugars are oxidized by the weakly alkaline copper reagent to a sugar acid, while Cu^2+^ is reduced to Cu^+^. Then, the arsenic molybdate reagent is used to regenerate Cu^2+^ ions, thereby reducing arsenic molybdate and producing a characteristic blue color [72,73].

#### 3.1.2. Fucose Content

The Dische or cysteine–sulfuric acid assay is carried out to quantify l-fucose content in hydrolyzed fucoidan solutions [74]. The test consists of mixing the fucoidan solution with diluted sulfuric acid (1:6). Then, the reaction mixture is incubated at 100 °C for a period, and the reaction is stopped by cooling in an ice bath. Thereafter, an aqueous l-cysteine solution is added, and the absorbance is measured at two wavelengths, namely, 396 and 430 nm. According to the difference of those two measurements, the possible interference of hexoses can be excluded [75]. However, algal polyphenols may interfere to a great extent in colorimetric fucose determination. Alternatively, as fucose is a neutral sugar, it can be determined using more sensitive methods, such as HPLC and GC after derivatization [28]. Details are provided in Section 4.3 with regard to the investigation of fucoidan monomeric composition.

#### 3.1.3. Fucoidan Content

The usual problem in the quantitative determination of fucoidan content is the absence of an appropriate standard. Commercial preparations may be insufficiently purified and may be structurally different from analytical samples. Nevertheless, on the basis of the anionic characteristic of fucoidans, thiazine dyes, such as in the toluidine blue (TB) assay according to Hahn et al. [76] and the Heparin Red^®^ Ultra assay according to Warttinger et al. [77,78], can be applied. The TB assay is based on the formation of a charge-transfer complex between the thiazine dye and the polysaccharide [79]. It consists of mixing fucoidan-containing solutions with TB at pH 1 for better reaction sensitivity. The absorbance is then measured at 632 nm using an aqueous solution of commercially purified fucoidan as a reference standard in a concentration range of 0–2.5 g·L^−1^. The color changes are demonstrated in Figure 3, whereby Figure 3A shows the metachromatic effect of fucoidan on the polycationic thiazine dye toluidine blue. A hypochromic effect is shown with a hypsochromic shift of the toluidine blue ultraviolet/visible light (UV/Vis) spectrum following the addition of polyanionic molecules (e.g., fucoidan). On the other hand, the Heparin Red^®^ Ultra assay is based on the fluorescence-quenching ability of fucoidans after incubation with Heparin Red^®^ reagent, as depicted in Figure 3C. It may be carried out using excitation and emission wavelengths of 570 and 605 nm, respectively. The reaction shows potential selectivity for fucoidan even in the presence of sodium alginate salt, as demonstrated in Figure 3D [78]. The Heparin Red^®^ Ultra assay also demonstrates great sensitivity in a linear range of 0.0–8.0 μg·mL^−1^. The results of such investigations indicate the relative quality of fucoidans and their degree of purity.

The principle behind the reaction of fucoidans with basic or cationic dyes was successfully applied using Alcian blue stain for the detection of fucoidans and its fragments after degradation experiments with fucoidanases in carbohydrate polyacrylamide gel electrophoresis (C-PAGE) [82]. Moreover, other similar anionic polysaccharides from carrageenan could be detected using the same principle [83]. Currently, several commercial highly purified fucoidans are marketed by well-known companies, such as Sigma-Aldrich^®^ and Marinova^®^, derived from *F. vesiculosus* and other brown algae species [17,84].

A more sensitive and selective electrochemical method for the detection of fucoidan was developed by Kim et al. in biological fluids and nutritional supplements. The method is based on potentiometric sensors using polyion-sensitive membrane electrodes. Examples of compounds acting as ion exchangers were tridodecyl methylammonium (TDMA) and dinonylnaphthalene sulfonate (DNNS) [17,85].

#### 3.1.4. Sulfate Content

As developed by Dodgson and Price, sulfate content can be analyzed on the basis of barium sulfate (BaSO_4_) precipitation after the addition of barium chloride (BaCl_2_) in gelatin using sodium sulfate (Na_2_SO_4_) or potassium sulfate (K_2_SO_4_) [86,87]. The sulfate amount is determined by turbidimetry at 500 nm [88]. Since sulfate ester groups are susceptible to hydrolysis, turbidimetric analysis requires preliminary liberation of the sulfate groups via acid hydrolysis using 4 M HCl at 100 °C for 6 h [89] or 2 M trifluoroacetic acid (TFA) at 100 °C for 8 h [90].

Using inductively coupled plasma mass spectrometry (ICP-MS), the sulfate content of fucoidan isolated from *L. hyperborean* was determined. Sulfur contents were determined by dissolving the dried fucoidan (70 °C for 90 min) in 1 M HNO_3_. The sulfation degree was determined by utilizing a mass balance equation, assuming that every sulfate group was associated with a sodium counterion [91].

#### 3.1.5. Uronic Acid Content

A colorimetric determination of uronic acids is usually performed using *meta*-phenylphenol according to the procedures presented by Filisetti-Cozzi and Carpita [15,92] or Blumenkrantz and Asboe-Hansen [93]. The same principle can be applied with *m*-phenylphenol to form a colored condensation product, where the sugar is firstly dehydrated by heating with sulfuric acid before the addition of *m*-phenylphenol and incubation at room temperature. The absorbance is then recorded at 525 nm. A modified uronic acid carbazole reaction is sometimes also applied [84,94].

Moreover, specific HPLC techniques based on monomer derivatization were reported. They include high-performance anion-exchange chromatography (HPAEC) coupled with pulsed amperometry detection (PAD). This method is commonly known as Dionex HPAEC–PAD, i.e., implementing a Dionex ICS-2500 system equipped with CarboPac™ PA20 analytical and guard columns. It depends on the fact that uronic acids are weak acids that can be derivatized to oxyanions at alkaline pH values [95,96].

In Section 3.2.3, alginate is discussed as a potential contaminant of fucoidans, leading to an increase in uronic acid content in fucoidan products if not properly removed. Hence, the identification of uronic acids is necessary to distinguish the components of fucoidan from the components of alginic acids. The uronic acids of fucoidans mainly constitute α-d-glucuronic acid [97,98], while those in alginate constitute α-l-guluronic acid (G-block) and β-d-mannuronic acid (M-block) linked via *α*-(1→4) bonds [99], as shown in Figure 4. These blocks produce a characteristic NMR pattern, from which the M/G ratio can be calculated [100,101].

### 3.2. Potential Coextracted Impurities

Since fucoidans are found in a highly complicated cell-wall matrix in addition to other polymers, such as cellulose, alginate, and protein, as well as polyphenols [102], several investigations should be carried out to detect and quantify such components. Moreover, other components may be also coextracted and present in crude fucoidans such as laminaran, mannitol, lipids, and pigments [59,103,104]. Hence, comprehensive downstream processes should be applied to remove all of these compounds as best as possible [54]. However, for reproducible and trusted biological activities, potential contaminants, such as proteins, alginate, laminaran, and total phenolic content should be quantified to determine the quality grade of fucoidans.

#### 3.2.1. Protein

The Folin–phenol [105] and Bradford assays are applied to determine protein content in fucoidan products, using bovine serum albumin as a reference standard for calibration [15,106]. The Lowry and Bradford assays are based on colorimetric determination, where they produce colored solutions recorded at 750 and 595 nm, respectively, in response to protein and/or amino acids. The Folin–phenol reagent consists of phosphomolybdic–phosphotungstic acid, which is reduced to a blue-colored solution by protein in an alkaline Cu^2+^ tartrate solution [105,107], whereas the color in the Bradford assay is formed due to complex formation between the protein and the Coomassie blue G-250 dye. Under acidic conditions, the protonated red dye is transformed to an anionic blue form through a dye–protein electrostatic and hydrophobic interaction [108,109].

Both assays show variable results, due to variations in protein composition, pH, and sample concentration [107], whereby only the tyrosine, tryptophan, and cysteine amino acids can react [110]. In addition, the Lowry method is not specific enough since the results are highly affected by the presence of interfering compounds that can also chelate Cu^2+^ (e.g., nitrogenous and phenolic compounds) [110,111].

#### 3.2.2. Phenolic Compounds

Phenolic compounds in brown algae vary structurally from simple molecules (e.g., hydroxybenzoic acid derivatives, such as gallic, phenolic, and cinnamic acids) or flavonoids (e.g., flavan-3-ol derivatives, such as epicatechin or epigallocatechin) to more complex phlorotannin polymeric structures (e.g., phlorethols, fuhalols, fucols, fucophlorethols, and eckol) [112].

As previously discussed, polyphenols are tightly noncovalently bound to fucoidans in the cell wall, which contribute along with fucoxanthin to the brown color of the crude fucoidan extract [54]. The total phenolic content can be quantified using the Folin–Ciocalteu method, especially for crude fucoidan products [113,114]. Additionally, the 2,4-dimethoxybenzaldehyde (DMBA) assay may be applied for phlorotannin content [115]. The Folin–Ciocalteu method is similar to the Folin–phenol applied for protein determination; however, the absorbance is recorded at 620 nm [114]. Nonetheless, interference from sugar monomers is common and may lead to false results. Gallic acid is commonly used as a reference standard and, therefore, the results are expressed as gallic acid equivalents [116].

#### 3.2.3. Alginate

Precipitation of alginate by divalent ions (e.g., Ca^2+^ or Ba^2+^) is a common pretreatment step during fucoidan extraction [35,117]. An acidic medium, i.e., below the *p*Ka of carboxylic groups, also helps in the precipitation of alginate as alginic acid [118]. Therefore, for the efficient removal of alginate during fucoidan extraction, both conditions are usually applied [38]. Nevertheless, traces of alginate are frequently detected in crude fucoidan extracts from brown algae [96]. Even the application of enzyme-assisted extraction employing an alginate lyase from *Sphingomonas* sp. (SALy) resulted in the crude fucoidan product containing substantial alginate, thus requiring a further purification step [119,120].

Since alginate is composed of *β*-d-mannuronic (M-block) and *α*-l-guluronic (G-block) acids as building blocks [121], it may interfere with the determination of uronic acids during fucoidan chemical characterization. Therefore, alginate can instead be determined as a function of the metachromatic change induced upon binding to cationic dyes, such as 1,9-dimethyl methylene blue (DMMB) [122], or using the TB assay. However, due to the different *p*Ka values of the sulfate ester group in fucoidans and carboxylic group in alginate, the different measurements at pH 1.0 and pH 7.0 can be used to quantify alginate content, where, at pH 1.0, fucoidan is ionized and interacts only with TB, while, at pH 7.0, both are ionized and induce color changes [76]. Dionex HPAEC–PAD can potentially be applied for the specific determination of alginate building blocks, thereby excluding interference from the uronic acids of fucoidans [96].

#### 3.2.4. Laminaran

Laminaran is a neutral water-soluble glucan found in brown algae functioning as a reserve food [103,104]. Its presence in crude fucoidan preparations is highly possible, owing to its precipitation with fucoidan after the addition of high volumes of ethanol (e.g., 70% *v/v*). Enzyme-assisted fucoidan extraction conducted using commercial enzyme mixtures, i.e., carbohydrase mixtures, can target the degradation of laminarin, leading to its removal [119]. Fortunately, laminarin cannot interact with cationic dyes during the determination of fucoidan content using TB and perylene diimide derivative (PDD) assays. The same principle is applied in the purification of fucoidan using anion exchange chromatography (e.g., DEAE–cellulose) in the presence of laminaran [54]. Therefore, laminaran is easily separated from fucoidan after the first step of purification.

## 4. Physicochemical Characteristics and Structural Features

### 4.1. Elemental Analysis

Elemental analysis is very important for comparative studies, which may be used to compare different fucoidan fractions as a tool to justify the purification process. A decrease in nitrogen content (%) and an increase in sulfur content (%) are critical elements for fucoidan quality, which may be interpreted as the removal/absence of proteins and an improvement in the sugar monomer–sulfate ratio, respectively [38,117,123]. Hence, protein content can be estimated by multiplying the percentage of N by 6.25. Similarly, the content of sulfate groups (as –SO_3_^−^Na^+^) can be calculated on the basis of the percentage of S [21,44,113]. Moreover, for the determination of sulfation degree in fucoidans, a number of equations were developed, as shown in Equations (1) and (2) [44].
(1)$$\mathrm{NSS}=\;\frac{C\%/12\;}{S\%/32}\;/\;6$$
(2)Degree of sulfation=1/NSS
where NSS is the number of sulfate esters per monosaccharide, 12 and 32 are the atomic weights of carbon and sulfur, respectively, and 6 is the number of carbon atoms in a sugar monomer assuming that all monomers in the polymer are hexoses.

### 4.2. Molecular Weight Averages

The molar mass, molecular size distribution, and chain conformation of polymers are among the important parameters affecting fucoidan applications. The molecular size of fucoidans ranges from 13 to 950 kDa. They can be classified, according to their molecular weight, into three classes: low-molecular-weight fucoidans (LMWFs) with a polymer size <10 kDa, medium-molecular-weight fucoidans (MMWFs) (10–10,000 kDa), and high-molecular-weight fucoidans (HMWFs) (>10,000 kDa) [124]. Chromatographic methods, i.e., gel filtration and anion exchange, are mostly used, whether separately or in combination, for the molecular mass determination of fucoidans. Examples of columns used include diethylaminoethyl (DEAE) Sepharose, Zorbax GF-450, OH-PAK SB-806 HQ, Sephadex G-50, Sephadex G-100, DEAE Toyopearl 650 M, Superdex 75 HR, and DEAE cellulose. The column should be firstly calibrated with polysaccharides of definite molecular sizes such as dextrans, pullulans, carrageenans, or heparins. However, the results show a range of molecular sizes instead of an exact value. Moreover, a polyacrylamide gel system, i.e., C-PAGE, can be used, where the polysaccharides are stained with a combination of Alcian blue and silver nitrate. C-PAGE can only be used to separate LMWFs in contrast to native unhydrolyzed polymers which are retained at the top of the gel [14,38,82].

Recently, with the aid of gel permeation or size-exclusion chromatography (GPC/SEC) coupled with multi-angle static light scattering, quasi-elastic light scattering and refractive index detection system (i.e., SEC–MALS–QELS–RI analysis), important structural characteristics could be estimated, in the context of Mw, Mn, Mp, PDI, and size (root-mean-square (rms) radius, *R_h_*) [125]. The number-average molecular weight or molar mass (Mn) indicates the average molecular weight of all polymer chains, while the weight-average molecular weight (Mw) considers the molecular weight of chains contributing to the molecular weight average. On the other hand, the molecular weight of the highest peak (Mp) determines the mode of molecular weight distribution. In addition, the polydispersity index (PDI) measures the broadness of the molecular weight distribution of polymers, where a larger PDI denotes a broader molecular weight distribution [126]. Equations (3)–(5) are applied for measurement of the above parameters, and they are now an integral part of many applications or as as add-on used in GPC/SEC techniques, such as the GPC Extension from Clarity Chromatography Station (starting from version 2.3).
(3)$$\mathrm{Mn}=\;\frac{\sum^{\;}N_{i}{\;M}_{i}}{\sum^{\;}N_{i}}$$
(4)$$\mathrm{Mw}=\frac{\sum^{\;}N_{i}M_{i}^{2}}{\sum^{\;}N_{i\;}M_{i}}$$
(5)$$\mathrm{PDI}=\;\frac{\mathrm{Mw}}{\mathrm{Mn}}$$
where Mi is the molecular weight of a chain, and Ni is the number of chains of that molecular weight.

Natural polymers such as proteins are usually monodisperse, with a PDI of approximately 1. However, polysaccharides are quite different, exhibiting PDIs greater than 1 [127]. The production of fucoidans with different molecular weight averages is possible through fractionation methods, such as via precipitation with increasing volumes of acetone, ethanol, or isopropanol, filtration membranes of a defined molecular weight cutoff (MWCO) [84], and gradient elution using NaCl during the elution step of ion-exchange chromatography purification, where a higher NaCl molarity elutes fucoidan fractions characterized by higher polarity, i.e., a higher sulfation degree and higher molecular weight [38,128].

### 4.3. Monomeric Composition

The detection of sugar monomers in polysaccharides is conducted after the step of acid hydrolysis [129], e.g., heating at 90–121 °C for 2–4 h with 4 M trifluoroacetic acid (TFA), applied for the isolation of sulfated galactofucan from the sporophyll of *U. pinnatifida* [130], 2 M HCl, applied for the commercial fucoidans of *F. vesiculosus* [88], and the two-step sulfuric acid treatment for 60 min with 72% H_2_SO_4_ at 30 °C followed by hydrolysis for 60 min in 4% H_2_SO_4_, applied for the crude fucoidans isolated from *Saccharina latissima* and *Laminaria digitata* [129,131]. Afterward, the hydrolysate is neutralized, filtered, and subjected to a liquid chromatography (HPLC) step, using different reference sugar monomers, such as arabinose, fructose, fucose, galactose, glucose, glucosamine, mannose, and xylose. The lead form Aminex HPX-87P column (Bio-Rad, Hercules, CA, USA) is widely used for carbohydrate analysis, operated at 80 °C and coupled with a refractive index (RI) detector [97,131]. A MetaCarb 67H column (Agilent Technologies, Santa Clara, CA, USA) operated at 45 °C may also be applied [132].

Moreover, the analysis of neutral and amino-containing monomers can be achieved via derivatization. The conversion of monomers to alditol acetate derivatives is more reliable, with subsequent separation using organic volatile solvents (e.g., dichloromethane), followed by analysis performed using GC/ESI-MS [8,133]. The reaction cascade consists of an initial reduction in alkaline medium followed by acetylation using acetic anhydride in the presence of 1-methylimidazole [84].

Recently, a comprehensive and fast method was developed by Rühmann et al. as function of monomer derivatization by 1-phenyl-3-methyl-5-pyrazolone (PMP) using liquid chromatography equipped with different detectors, such as UV and MS detectors or both (LC–UV–ESI-MS/MS) [134], which was successfully applied for the characterization of fucoidan isolated from *F. vesiculosus* [38].

### 4.4. Glycosidic Linkage

Methylation using the CH_3_I/NaOH method for deacetylated and desulfated fucoidans, followed by hydrolysis, reduction, and acetylation as before, enables sugar derivatization to alditol acetates that can be determined using GC/MS [135]. The scheme of reactions is summarized in Figure 5. Complete methylation can be confirmed upon disappearance of the OH band (3200–3700 cm^−1^) in the IR spectrum [136]. This method is commonly performed to determine glycosidic linkages and, consequently, branching sites in polysaccharides.

The desulfation step can be carried out enzymatically using sulfoesterase or chemically. Despite the regioselectivity of the enzymatic reaction, chemical desulfation is frequently conducted [60]. The chemical reaction involves a solvolytic desulfation, which consists of firstly passing the fucoidan solution through a cationic exchange resin, i.e., Amberlite CG-120 column (H^+^-form). Then, the main desulfation step is conducted via incubation of the pyridinium salt of fucoidan in dimethyl sulfoxide (DMSO) at 100 °C for 3–10 h [137,138,139]. In addition, the incubation of equal volumes of desulfated fucoidan with concentrated aqueous NH_3_ overnight at elevated temperature, i.e., 37 °C, resulted in the deacetylation of fucoidans [33,139].

Alternatively, periodate oxidation/Smith degradation can be applied for the analysis of fucoidan glycosidic linkages. The fucoidan is oxidized with NaOI_4_ and then reduced with NaBH_4_. Afterward, the product is hydrolyzed using acid and analyzed via GC [140,141]. However, it is worth mentioning that the (1→3) linkages between fucose residues and the high degree of substitution of hydroxyls make fucoidans resistant to Smith degradation [33]. The analysis of glycosidic linkages can also be achieved using NMR, as further discussed in Section 5.2.

### 4.5. Others

Other characteristics, such as solubility and optical activity, should also be investigated [38]. It is well known that fucoidans are polar compounds and freely soluble in water. Hence, turbidity may indicate the presence of impurities, such as proteins and the Ca salt of alginate. In addition to water, fucoidans are highly soluble in solvents with high dielectric constants [59], as well as with acidic and alkaline pH values. However, to guard against polymer hydrolysis, their stability was investigated, showing that solutions prepared in a pH range from 5.8 to 9.5 were stable [142]. In contrast, fucoidans are practically insoluble in ethanol or cetyltrimethylammonium bromide (CTAB) and, therefore, both solvents can be applied for fucoidan precipitation and isolation form algal crude extracts [37,54].

Due to the predominance of l-(−)fucose in fucoidan backbones, fucoidans are optically active molecules, with fucoidan solutions showing a levorotatory characteristic when exposed to plane-polarized light [6]. Moreover, researches of fucoidan chemistry always try to link its chemical properties, i.e., molecular weight, polydispersity, branching, sulfate content, and uronic acid content, with its physical rheological properties [133]. Fucoidan aqueous solutions generally possess low viscosity. Nevertheless, the degree of viscosity mainly depends on the temperature, pH, molecular weight, concentration, sulfate content, and degree of branching [142,143,144,145]. One example is fucoidan isolated from *L. japonica*, which has a molecular weight of 10.5 kDa and a high content of fucose and sulfate. Increasing the fucoidan concentration resulted in an increase in solution viscosity. Meanwhile, increasing fucoidan concentration led to a decrease in solution pH [146]. In addition, the aqueous solutions of fucoidan isolated from *L. religiosa* and *U. pinnatifida* showed low viscosity, characterized by a pseudoplastic rheological behavior [144]. Furthermore, Monsur et al. investigated several fucoidan fractions isolated from *Turbinaria turbinata*. They were Newtonian fluids regarding the direct relationship between shear stress and shear rate, as well as in the context of solution concentration. Concurrently, they found that the fraction with the highest molecular weight, sulfate content, and polydispersity exhibited the lowest viscosity [133].

Furthermore, the type of algal species and the presence of ions can affect the viscosity of fucoidan solutions [146]. The highest viscosity of a fucoidan aqueous solution was reported for fucoidan isolated from *F. vesiculosus* algae species [147], while the viscosity of fucoidan solutions obtained from *Cladosiphom okamuranus* increased linearly with increasing polymer concentration up to 2% (*w/v*), as well as after the addition of salts such as NaCl or CaCl_2_. However, viscosity may reflect the amount of contaminating alginate left in the fucoidan extract, since the viscosity of alginate is very high [148,149].

Interestingly, fucoidans differ from other polysaccharides, as they do not possess the ability to form a gel alone, where mixing with other positively charged polymers is needed to produce gel, owing to ionic interactions [145,146]. The thermal degradation of fucoidans, including their melting points, was studied [38,133]. For further information on other physical parameters such as the consistency, flow behavior, and rheological properties of fucoidans, as well as the ways in which these properties affect fucoidan applications in the pharmaceutical industry, including drug delivery, interested readers may refer to [146,150], where the design of fucoidan-based nanosystems and other nanocarriers encapsulating fucoidan was shown to depend on the physiochemical behavior of fucoidans, e.g., physical appearance, chemical features, molecular weight (Mw), solubility, pH, and melting point [146,151].

## 5. Spectrometry and Chemical Characterization

Due to their complex chemical structures, several spectrometric methods (e.g., FT-IR, NMR, and MS) have been used to elucidate the structural features of fucoidans, including the position of sulfate groups and glycosidic bonds, and the molecular weight. Furthermore, the application of regio- and stereoselective fucoidan-degrading enzymes isolated from marine bacteria provided new insights into the chemical structure of fucoidan [43,74].

### 5.1. FT-IR

The preliminary identification of fucoidan functional groups is always performed by scanning samples using FT-IR between 400 and 4000 cm^−1^ [152,153]. Fucoidans show characteristic and typical IR bands for their functional structural building blocks (e.g., the O–H group of monomeric monosaccharides, C–H, asymmetric stretching of S=O and C–O–S of sulfate ester groups, and O–C–O and C–O–C of glycosidic and intramolecular linkages at 3421, 2940, 1221, 827, 1634, and 1010 cm^−1^, respectively) [123,153]. Moreover, peaks between 1650 and 1800 cm^−1^ for C=O groups indicate whether fucoidans are acetylated and contain uronic acid residues [133,154,155].

Other important information may be extracted from the IR spectra, such as the axial position of the sulfate ester group at C-4 [21,156]. A complex pattern between 840 and 800 cm^−1^ is commonly shown, indicating the different substitutions of sulfate ester groups at the most abundant C-4 and C-2/C-3 positions, showing a peak and a shoulder for the axial 4-position of C–O–S and equatorial 2/3-position of C–O–S, respectively [157]. To exclude the bending vibration of the C–H group of sugars, IR results should be compared with NMR data after sulfate ester alkali hydrolysis and methylation analysis to determine the exact sulfate position [6,158]. In addition, IR bands at 622 and 583 cm^−1^ result from the asymmetric and symmetric O=S=O deformation of sulfates [159].

Anomeric C–H deformation may be also identified in IR spectra, where the *β*-anomeric type is represented by a small peak at 890 cm^−1^, while the *α*-analogue appears theoretically at around 860 cm^−1^, which may be overlapped by stronger sulfate bands in the same range [8,140].

However, the structural information provided in FT-IR spectra may not be very valuable, especially for the determination of secondary sulfate ester groups. Signals describing the positions of secondary sulfate groups depend on the real conformation of monosaccharide units, which may be considerably distorted in branched and heavily sulfated chains by neighboring substituents [160].

### 5.2. NMR

Previously published articles have shown the valuable information presented by NMR for structural elucidation and the numerous structural features of different fucoidans from different origins. One- and two-dimensional (1D and 2D) NMR experiments of fucoidans, including ^1^H- and ^13^C-NMR, demonstrate relatively well-interpreted spectra related to the sulfation pattern and glycosidic linkages. Two approaches are usually applied in NMR experiments for obtaining valuable structure information, i.e., careful fractionation of crude fucoidan and/or specific chemical modification, such as desulfation. Such treatments are aimed at producing regular or masked regular backbones, resulting in NMR spectra that can be interpreted [161]. An example of successful fractionation of a crude algal fucoidan to obtain a fucoidan fraction having regular structure, which was elucidated using NMR spectra, was described by Bilan et al., where they obtained a regular fucoidan from the brown seaweed *Fucus distichus* [162]. Fractionation is commonly conducted during the elution of purified fucoidan from the anion-exchange column via the salting out mechanism using different molar concentrations of NaCl, i.e., gradient elution. A relationship was identified between NaCl concentration and the molecular weight and sulfate content of the obtained fucoidan fraction, thereby facilitating the structural elucidation of various relatively simple fractions [38,119]. Moreover, fractionation may also carried out through dialysis membranes of different molecular weight cutoff (MWCO), such as the fractionation of crude fucoidan into LMWFs and HMWFs [28]. For further information on such treatments, interested readers can refer to the recently published review discussing the different downstream processes applied in fucoidan production by Zayed et al. [54]. The production of oligomer fractions or fragments is considered a potential tool for polymer simplification, i.e., enzymatic depolymerization, prior to NMR experiments. This step can be performed by enzymatic treatment of the native fucoidan with fucoidan-degrading enzymes, i.e., fucoidanases [161,163]. In addition, the application of fucoidanases on desulfated or deacetylated fucoidans may result in more valuable structural information [34]. The desulfation and deacetylation of fucoidans is applied to produce simpler compounds, allowing a comparison of the chemical shifts (δ, ppm) with standard sugars or the native polymer, especially because sulfate ester groups cause deshielding of neighboring protons and carbons, consequently appearing downfield in the NMR spectra [8,139,163,164]. Some examples of fucoidans or sulfated fucans elucidated by 1D and 2D NMR are discussed below in detail.

#### 5.2.1. 1D NMR

Fucoidans from *F. vesiculosus* and *Ascophyllum nodosum* (Fucales) are the simplest form of algal fucoidans. They are polymers of alternating *α*-(1→3)- and *α*-(1→4)-linked l-fucopyranoside repeating units ([→4)-*α*-l-Fuc*p*-(1→3)-*α*-l-Fuc*p*-(1→4)-*α*-l-Fuc*p*-(1→3)-*α*-l-Fuc*p*(1→]) [35,43], where some structural features can be elucidated from NMR spectra. In the ^1^H-NMR spectrum, singlet peaks of the shielded protons at around 1.2 ppm are assigned to the –CH_3_ groups (H-6) of the l-fucose monomer. Other peaks appearing slightly shifted between 3.8 and 4.5 ppm can be assigned to H-2, H-3, H-4, and H-5. Moreover, the anomeric proton H-1 can be observed deshielded at 5.2 ppm, confirming the *α*-linked sugar monomers. The presence of other sugars such as galactose, mannose, and xylose can be deduced from signals in regions lower than 3.7 ppm, as reported in the fucoidan isolated from *S. polycystum* [165]. The NMR analysis of these fucoidans was recorded for their native forms, which produced low-resolution spectra.

Additionally, in the ^13^C-NMR spectrum of a fucoidan fraction isolated from *S. mcclurei* (Fucales) of a galactofucan nature, major intense peaks could be easily elucidated. They included the –CH_3_ group (C-6) appearing in an upfield region, i.e., 15–17 ppm, and the anomeric carbon at C-1 in a downfield region, i.e., 96–100 ppm. Other peaks showed several degrees of multiplicity owing to the presence of various glycosidic linkages and sulfation patterns [138]. These peaks are typical for an *α*-fucopyranoside backbone [166]. Acetylated moieties in the fucoidan structure can also be easily detected in ^13^C-NMR spectra, where methyl (–CH_3_) and carbonyl (C=O) groups of the *O*-acetyl group appear in upfield (i.e., 21–22 ppm) and downfield (i.e., 170–180 ppm) regions, respectively. Examples include fucoidans isolated from *S. japonica* and *U. pinnatifida* [167,168].

Furthermore, the effect of the sulfate ester group (–OSO_3_^−^) on the chemical shifts (δ, ppm) of attached C and accompanied Hs is among the features that can be revealed by NMR. This helps the elucidation of sulfation pattern with fucoidan backbones. An example includes the structural characterization of fucoidans isolated from *S. myriocystum*. The presence of a downfield signal at 4.84 ppm in ^1^H-NMR indicated a sulfate group at position C-4, while the chemical shifts of other hydrogen atoms C-2, C-3, and C-5 appeared at 4.80, 4.22, and 4.77, respectively. These data were confirmed in the ^13^C-NMR spectrum, where C-4 appeared at 81.33 ppm, while the C-3 position was confirmed by the signal at 76.39 ppm [169]. Similarly, *Nemacystus decipiens* fucoidan showed sulfation substitution at C-4 and, consequently, H-4 appeared more deshielded at 4.9 ppm than the other protons [170]. Furthermore, fucoidan, i.e., xylogalactofucan, isolated from *Sphacelaria indica*, showed 4-*O*-sulfated residues at C-4 according to the presence of a peak in the ^1^H-NMR spectrum at 4.4 ppm, which was assigned to H-4 and confirmed by IR analysis [171]. In other cases, the sulfation pattern was demonstrated and confirmed at mainly C-2 and partially C-4 in fucoidan isolated from *F. evanescens* C.Ag by NMR data [166]. However, other studies preferred analyzing desulfated and deacetylated fucoidan residues for simpler and more easily interpreted spectra [172].

The main l-fucopyranose backbone of fucoidan isolated from *S. binderi* was determined by comparing the ^1^H-NMR and attached proton test (APT) NMR spectra with commercial food-grade fucoidan and an *α*-l-fucose standard. The positions of the sulfate groups were determined through the difference in proton and ^13^C-NMR chemical shifts with respect to the *α*-l-fucose standard. The downfield proton and C chemical shifts at H-2 (0.90 ppm difference) and C-2 (14.97 ppm difference) compared with the peaks of *α*-_L_-fucose were due to the presence of a sulfate group at C-2 [173].

#### 5.2.2. 2D NMR

Homonuclear (e.g., correlation spectroscopy (COSY), total correlated spectroscopy (TOCSY), nuclear Overhauser effect spectroscopy (NOESY), and rotating frame Overhauser effect spectroscopy (ROESY)) and heteronuclear (e.g., heteronuclear multiple bond correlation (HMBC) and heteronuclear single quantum coherence (HSCQ)) 2d NMR techniques have been used to further reveal the potential structural secrets of numerous fucoidans (e.g., *F. serratus*, *S. latissimi*, *Chorda filum*, and *F. evanescens*) [139,167,172,174].

It is valuable to study the desulfated product of native fucoidans to identify the glycosidic linkages of their backbone, as in the case of fucan sulfate isolated from the *Holothuria albiventer* sea cucumber [174]. The correlation between H-1 and H-3 of polymer residues was identified in the ROESY spectrum, confirming the presence of (1→3) linkages. Furthermore, the sequences of the repeating units were also determined as a function of the long-range scalar HMBC correlations. For estimation of the configurations at the glycosidic linkages, the direct coupling constant (^1^*J*_C–H_) of C-1 for each monosaccharide residue was also obtained from the 2D ^13^C/^1^H HMBC spectrum. The large values of 170–175 Hz for these fucose residues indicated that the protons are equatorially positioned [175]. Moreover, taking account of the vicinal coupling constant (^3^*J*_1H–2H_) of 3 Hz for fucose residues, the configuration at C-1 of these residues was determined as the *α*-form [174].

Other highly regular homogeneous sulfated fucans were isolated from sea cucumbers *Holothuria fuscopunctata*, *Thelenota ananas*, and *Stichopus horrens*. Their glycosidic linkage sequences were obtained from the selected ROESY and HMBC cross-signals. The correlation between H-1 and H-4 of both *H. fuscopunctata* and *T. ananas* fucan sulfates was identified in the ROESY and HMBC spectra, whereas, for the *S. horrens* fucan sulfate, correlation between H-1 and H-3 was observed, confirming the presence of *α*-(1→4) glycosidic bonds in both *H. fuscopunctata* and *T. ananas* fucan sulfates. The structural sequences of the three fucoidans from *H. fuscopunctata*, *T. ananas*, and *S. horrens* were →4-*α*-l-Fuc*p*-(3SO_3_^−^)-1→, →4-*α*-l-Fuc*p*-(2SO_3_^−^)-1→, and →3-*α*-l-Fuc*p*-(2SO_3_^−^)-1→, respectively [176].

### 5.3. Mass Spectrometry

Mass spectrometry (MS) is widely used to provide valuable structural information, especially if the fucoidans vary in terms of their glycosidic linkages and/or sulfation patterns [138]. However, it is difficult to be applied to fucoidans with high molecular weights or a high degree of sulfation [163], since depolymerization or autohydrolysis of the polymer should firstly be carried out [177,178]. In addition, the labile nature of sulfate ester groups in high-molecular-weight fucoidans results in the polymers exhibiting desulfation rather than ionization [179]. Thus, the ions generated from desulfation dominate the mass spectra, limiting the obtained structural data of native fucoidans [125,180].

Therefore, prior to the mass spectrometric measurement, pretreatments are applied, including fucoidan depolymerization, leading to the generation of oligomers. In addition, structural modifications, including desulfation and deacetylation, are applied. Such pretreatments result in more easily interpreted spectra compared to the native high-molecular-weight sulfated forms. The polymer can be chemically depolymerized via partial acid hydrolysis using 0.2 N TFA at 60 °C or 0.75 mM H_2_SO_4_ at 60 °C following the solvolytic desulfation step [179]. In contrast with autohydrolysis, enzymatic depolymerization by fucoidanses was reported, albeit not widely [181]. Nevertheless, the role of fucoidan-degrading enzymes or fucoidanses is discussed in Section 6 regarding their enzymatic modifications of fucoidans as a potential prerequisite for conducting spectrometric analysis.

The analysis of oligomers can be performed using negative ion tandem electrospray ionization (ESI-MS) and matrix-assisted laser desorption/ionization (MALDI-TOF) mass spectrometers [135]. The reasonable interpretation of fragmentation patterns can reveal many structural features in the produced oligomers useful in understanding the structure of the whole polymer. The MALDI-TOF analyzer is more convenient for the analysis of fucoidan fragments, i.e., fucooligosaccharides, adding more sensitivity and accuracy with respect to classical MS and NMR methods which detect minor constituents, as in the case of fucoidan isolated from *F. evanescens* [179,182]. The systematic nomenclature developed by Domon et al. is still approved for carbohydrate fragmentations in the MS/MS spectra of glycoconjugates [183].

Anastyuk et al. reported an optimization protocol for the application of tandem MS techniques, including ESI-MS and MALDI-TOF-MS in the structural elucidation of fucoidans isolated from different brown algal species. The protocol also reported the common fragments of *S. cichorioides* fucoidan detected in MS spectra using both techniques in a comparative approach following autohydrolysis. Fragment ions at *m/z* 97 and 225 were always detected, representing the sulfate anion and [FucSO_3_–H_2_O], respectively. In addition, the *m/z* peaks at 243.02 and 389.08 corresponded to the [FucSO_3_]^−^ and [Fuc_2_SO_3_]^−^ fragment ions [179]. However, other characteristic fragments were detected at different *m/z*, due to differences related to ionization mode, including the inclusion of Na with the produced fragments, i.e., [Fuc_3_(SO_3_)_3_]^3−^ and [Fuc_3_(SO_3_Na)_3_–Na]^−^ at *m*/*z* 231.01 and 739.1 in ESI-MS and MALDI-TOF-MS spectra, respectively. The interpretation of such spectra confirmed the presence of 3-linked 2,4-disulfated *α*-l-fucan as the main backbone of *S. cichorioides* and *F. evanescens* fucoidans [179]. Cuong et al. elucidated the chemical structure of *Sargassum henslowianum* on the basis of MS data. The ESI-MS results demonstrated a major signal at *m/z* 243 corresponding to [M − H]^−^ of the monosulfated fucose [FucSO_3_]^−^ ion. Other signals indicated α-(1→3)-linked l-fucopyranose as the fucoidan backbone, whereas the sulfate positions varied, but were mostly located at positions C-2 (*m/z* 139), C-3 (*m/z* 169), and C-4 (*m/z* 183) of the fucose residues. These results depicted the sulfation pattern of this fucoidan to be mostly at C-2 and C-4 and sometimes at the C-3 position of fucose residues [90], according to the previous findings of Tissot et al. [184]. They showed that the three isomers have different fragmentation patterns. While the 3-*O*-sulfated fucose lost a hydrogenosulfate anion, the other isomers, i.e., 2-*O*-, and 4-*O*-sulfated fucose, exhibited cross-ring fragmentation, producing ^0,2^X and ^0,2^A daughter ions, respectively [184].

Sample preparation is also important, whereby the sample should be mixed with 0.5 M 2,5-dihydroxy benzoic acid (DHB) in MeOH for positive-ion experiments or 0.5 M arabinoosazone matrix for negative-ion experiments in acetone. The negative-ion experiment requires a further 10-fold dilution of the sample in water [185].

Galmero et al. recently established a new fast method for carbohydrate glycosidic linkage determination. The developed method employed ultrahigh-performance liquid chromatography coupled with triple-quadrupole mass spectrometry (UHPLC/QqQ-MS) analysis performed in multiple reaction monitoring (MRM) mode, using a library of 22 glycosidic linkages built from commercial oligosaccharide standards. Permethylation and hydrolysis conditions alongside LC–MS/MS parameters were optimized, resulting in a workflow requiring only 50 μg of substrate for analysis [91]. This method can be used in the future for the determination of fucoidan glycosidic linkages.

## 6. Role of Fucoidan-Degrading Enzymes in Structural Elucidation

Fucoidan-degrading enzymes can be applied for the production of chemically defined bioactive oligosaccharides by hydrolyzing the sulfated fucans and fucoidans [14,186,187]. They include fucoidanases and sulfatases, which play very critical roles in the structural elucidation of fucoidan macromolecules in combination with spectrometric methods [188,189,190]. They are characterized using mild conditions, as a function of regio- and stereoselectivity, to determine the exact pattern of sulfation and glycosidic linkages compared with toxic (e.g., pyridine used in solvolytic desulfation), nonselective, and tedious chemical or physical modifications [139,167,191,192]. Moreover, fucoidanases preserve the sulfation pattern, which is among the major factors implicated in various biological activities [130]. Such enzymes are mainly isolated from symbionts (e.g., Proteobacteria and Bacteroidetes) associated with brown algae or marine invertebrates [193]. Interestingly, Ohshiro et al. detected both activities, i.e., fucoidan desulfation and depolymerization, using degrading enzymes isolated from the *Flavobacterium* sp. F-31. These enzymes worked on the fucoidan isolated from *Cladosiphon okamuranus* as a carbon source. Nevertheless, the desulfation activity of such enzymes was notably detected following the enzymatic degradation step of native fucoidan, i.e., depolymerized fractions [194].

Fucoidanases were first described in 1967 by Thanassi and Nakada after their isolation from the hepatopancreas *Haliotus* sp. [34,195]. They are among glycosidases (EC3.2.1.-GH 107) that catalyze the hydrolysis of glycosidic bonds between sulfated fucose residues in fucoidans [9,195]. On the basis of their mode of action, they were previously classified according to the types of glycosidic linkages they act on and further subclassified as *endo*- or *exo*-hydrolyases. Therefore, they were recently classified as *endo*-fucoidanases or α-l-fucoidan endohydrolases (EC3.2.1.44 or EC3.2.1.211 and EC3.2.1.212), as defined in Expasy, where EC 3.2.1.211 is believed to cleave *endo*-α-(1→3) l-fucoside linkages, while EC 3.2.1.212 likely cleaves *α*-(1→4) l-fucoside linkages, in addition to the *exo*-type, including EC 3.2.1.B47, without affecting the sulfate ester groups in the fucoidan backbone. In contrast, there are fucosidases which catalyze the cleavage of nonsulfated fucose residues from other fucose-containing compounds [195,196]. Moreover, fucoglucoronnomannan lyases were first shown to cleave linkages between mannose and glucuronic acids in a lyase manner by Takayama et al. in 2002 [197] and Sakai et al. in 2003 [198]. Later, they were analyzed on different fucoidan substrates by Cao et al. in 2018 [9] and suggested to cleave *α*-(1→4) linkages. They include the *endo*-fucoglucuronomannan lyases FdlA (GenBank accession number: AAO00510.1) and FdlB (GenBank accession number: AAO00511.1) [9].

In contrast, not many sequences of sulfatases acting on fucoidans have yet been found [199], as fucoidan sulfatases or sulfoesterases have not yet been well explored. Scarce studies have involved their putative mechanisms in the investigation of fucoidan sulfation pattern. They include enzymes isolated from the marine bacterium *Wenyingzhuangia fucanilytica* named SWF1 (GenBank accession number: WP_068825883.1) and SWF4 (GenBank accession number: WP_068828765.1) belonging to the family of formylglycine-dependent enzymes (SulfAtlas), i.e., *exo*-2*O*- and -3*O*-fucoidan sulfatase) [190] and *Pecten maximus*, i.e., 2*O*- fucoidan sulfatase [137].

Marine bacteria and mollusks have been reported to be the major sources of fucoidanases [9]. Among frequently used fucoidanases is FcnA (GenBank accession number: CAI47003.1) and its *C*-terminal truncated version named FcnA2. FcnA was produced by recombinant DNA technology via cloning its encoding gene from the marine bacterium *Mariniflexile fucanivorans* SW5T. FcnA2 showed *endo* α-(1→4) cleavage activity on *Pelvetia canaliculate* fucoidan [200]. Hence, the recent production of modified stabilized fucoidanases, i.e., FcnA2, via C-terminal target truncation helped the enzymatic production of more defined fucoidan fractions from different fucoidans [9]. In addition, Fda1 (GenBank accession number: AAO00508.1) and Fda2 (GenBank accession number: AAO00509.1) produced by *Alteromonas* sp. SN-1009 were α-(1→3)-specific, catalyzing the cleavage of the fucoidan isolated from *Kjellmaniella crassifolia* or *Saccharina sculpera* [197]. Moreover, the marine bacterium *Formosa algae* (KMM 3553T) produced FFA1 (GenBank accession number: WP_057784217.1) and FFA2 (GenBank accession number: WP_057784219.1). They demonstrated *endo α-*(1→4) cleavage activity on fucoidan isolated from *F. evanescens* [161]. FcnA, Fda1, Fda2, FFA1, and FFA2 all belong to the glycoside hydrolase family GH107 in CAZy [16,196,201]. Recently, a novel *endo*-fucoidanse was characterized and recognized as the first member of the GH168 family in CAZy. It was isolated from the marine bacterium *W. fucanilytica* CZ1127^T^ and encoded by the gene *funA*. Heterologous expression of the gene resulted in the production of FunA that specifically cleaved the α-(1→3) glycosidic linkage between the 2-*O*-sulfated and nonsulfated fucose residues of the sea cucumber *Isostichopus badionotus* sulfated fucan [202].

In the last few years, several sulfated fucan hydrolases were isolated. However, most studies were aimed at characterizing enzyme specificity and decreasing the molecular weight of fucoidans for modification of their activities and easy handling, whereas only few studies produced fragments which were structurally defined by spectrometric analysis. Examples include fucoidanases isolated from *Luteolibacter Algae* H18 and *Flavibacterium* sp. F-31 catalyzing the hydrolysis of *C. okamuranus* fucoidans [194,203,204], the gut contents of the sea cucumber *Sticopus japonicus* (Strain, SI-1234) catalyzing *C. okamuranus* and *A. nodosum* fucoidans [198], *Sphingomona spaucimobilis* PF-1 (FNase S) catalyzing *U. pinnatifida* fucoidans [130], Alteromonadaceae (Strain SN-1009) catalyzing *Kjellmaniella crassifolia* fucoidans [205], *Dendryphiellaarenaria* TM94 catalyzing *F. vesiculosus* fucoidans [206], and *Pseudoalteromonas citrea* KMM 3296 and *Littorina kurila* catalyzing *F. evanescens* fucoidans [207]. In addition, a novel *α*-l-fucosidase was isolated from the marine bacterium *Wenyingzhuangia fucanilytica* CZ1127^T^ that acts on the *α*-(1→4)-fucosidic linkage of *Thelenotaananas* (wild sea cucumber) fucoidans [208]. Table 1 summarizes the previously characterized fucoidanases and fucoidan sulfatases that produced well-defined fucoidan fragments.

## 7. Conclusions and Future Perspectives

Fucoidans are multifunctional macromolecules, in which several structural motifs participate in and affect their wide spectrum of applications. Therefore, numerous characteristics should be assigned (e.g., sulfation pattern and content, monomeric composition, degree of purity, molecular weight distribution, and glycosidic linkage) before their further applications, especially as there are several marketed fucoidan products. Hence, single analytical methods are not able to answer all the questions regarding the various fucoidan features. It is also beneficial if the fucoidan characteristics can be related to the required activities. This may help researchers to understand and reveal some of fucoidan’s secrets. Somewhat old techniques are still applied; however, recently, other novel techniques were developed to more easily obtain and confirm results. These include advanced spectrometric (i.e., NMR and MS) and chromatographic coupled to spectrometric (GC/MS) methods. Moreover, fucoidan-degrading enzymes play a potential role prior to further analyses with advanced spectrometric methods, and they possess several advantages compared with chemical modification methods.

Therefore, the outlook in the field of fucoidan characterization and structural elucidation should focus on the metagenomic analysis of genes encoding such enzymes, where their overexpression and characterization may be a more effective tool in combination with spectrometric techniques. This will enable an understanding of the fucoidan mechanism of action and its exact interaction with different human targets.

## Figures and Tables

**Figure 1 marinedrugs-18-00571-f001:**
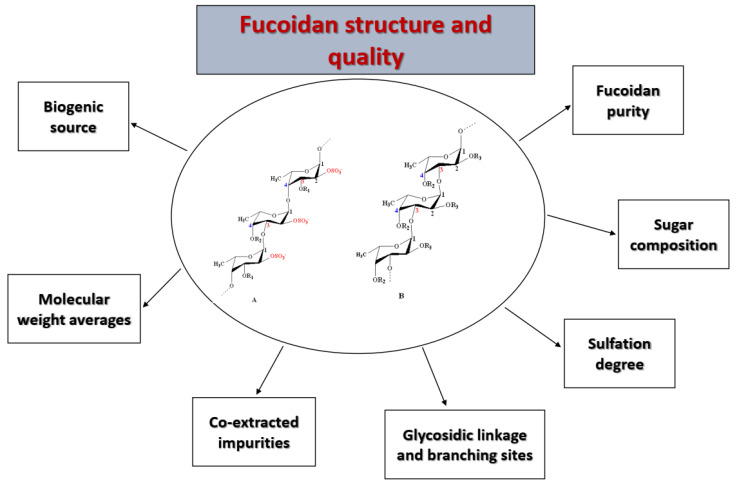
Summary of the different factors affecting the biological behavior of fucoidans. These factors should be characterized for the successful application of fucoidans.

**Figure 2 marinedrugs-18-00571-f002:**
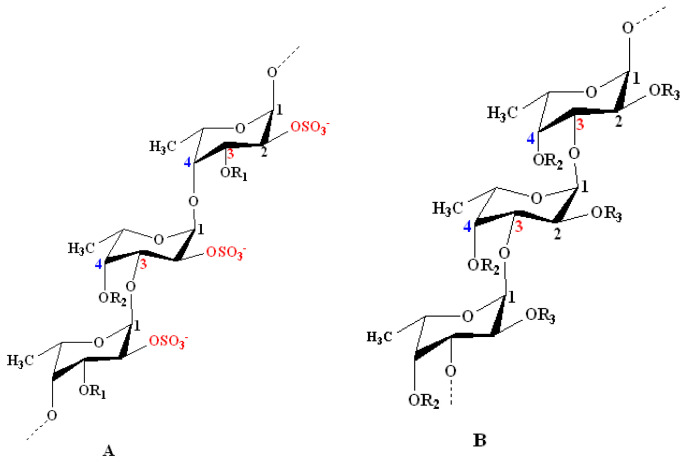
Structural models for the chemical structure of fucoidans derived from some species of seaweeds as proposed by Cumashi et al. and Ale et al. [35,64]. **Model A**: Model representing fucoidans from some species of Fucales. It shows repeating l-fucopyranoside units linked with alternating *α*-(1→4) and *α*-(1→3) glycosidic linkages. C-2 is always substituted with a sulfate ester group. Examples include *Fucus vesiculosus* and *Ascophyllum nodosum*: R_1_ = SO_3_^−^, R_2_ = H; *F. serratus* L.: R_1_ = H, R_2_ = side chain or SO_3_^−^; and *Fucus evanescens* C. Ag: R_1_ = H, R_2_ = SO_3_^−^ or H. **Model B**: Model representing some species of Laminariales and Chordariales. Both orders show a repeated *α*-(1→3)-linked branched l-fucopyranoside backbone at C-2. Sulfate ester groups mainly substitute C-4 and sometimes C-2. Examples include *Laminaria saccharina* (Laminariales): R_2_ = OSO_3_^−^, R_3_ = H alternating with OSO_3_^−^ and l-fucose; and *Cladosiphon okamuranus* (Chordariales): R_2_ = OSO_3_^−^ alternating with H, R_3_ = OSO_3_^−^ alternating with H and uronic acid. Other minor sugar units (e.g., mannose and galactose) and acetyl groups occur in fucoidan structures at certain unknown positions [64].

**Figure 3 marinedrugs-18-00571-f003:**
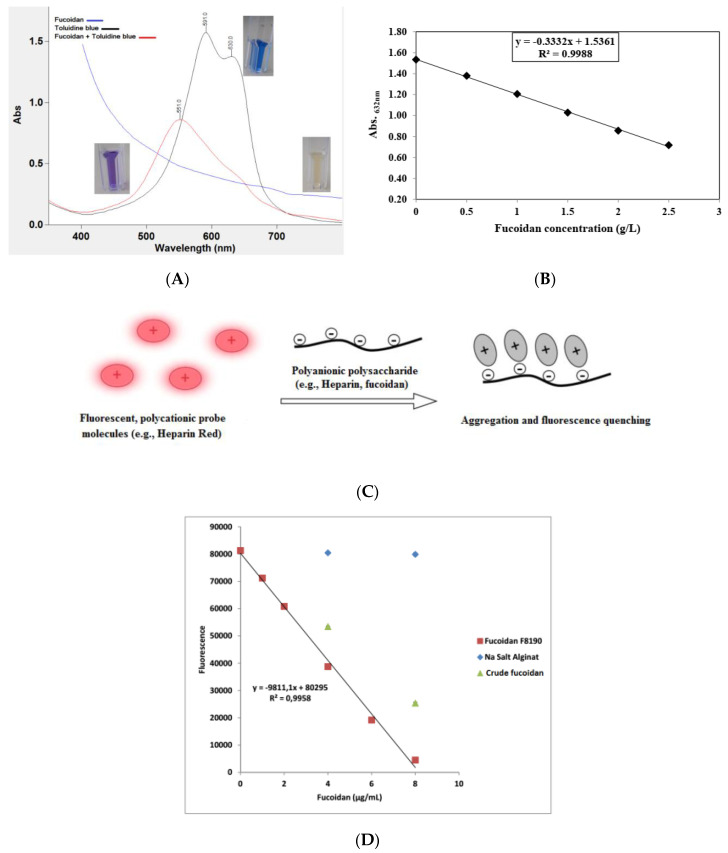
(**A**) Metachromatic effect of fucoidan on the polycationic thiazine dye toluidine blue (TB). A hypsochromic shift and hypochromic effect are observed after the reaction of TB with fucoidans. (**B**) Calibration curve of TB assay showing the reaction linearity in a specified fucoidan concentration, i.e., 0–2.5 g·L^−1^. (**C**) Representation of polyanionic polysaccharide reaction with fluorescent perylene diimide molecules (e.g., Heparin Red^®^). The reaction electrostatically produces aggregates, followed by fluorescence quenching (modified according to [80]). (**D**) Calibration curve of Heparin Red^®^ assay showing crude fucoidan samples deviating from the linear range of the reference sample with no interference from alginate [81]. The ultraviolet/visible light (UV/Vis) measurement was conducted using a UV/Vis spectrometer (Cary 60 UV/Vis, Agilent Technologies, USA), while the fluorescence was recorded using a spectrofluorometer (FP-8300, JASCO Deutschland GmbH, Germany).

**Figure 4 marinedrugs-18-00571-f004:**
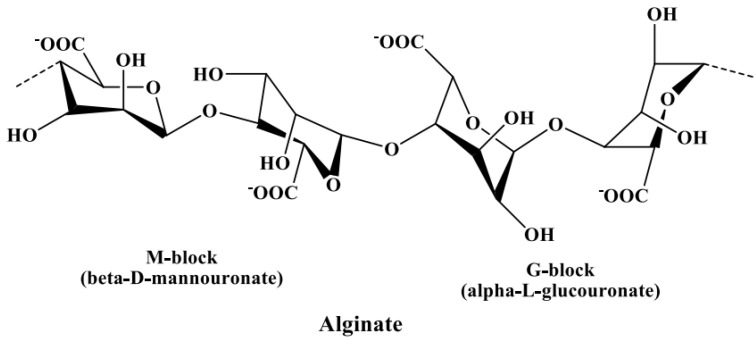
Chemical structure of alginate composed of α-l-guluronic acid (G-block) and β-d-mannuronic acid (M-block) linked via *α*-(1→4) glycosidic bonds.

**Figure 5 marinedrugs-18-00571-f005:**
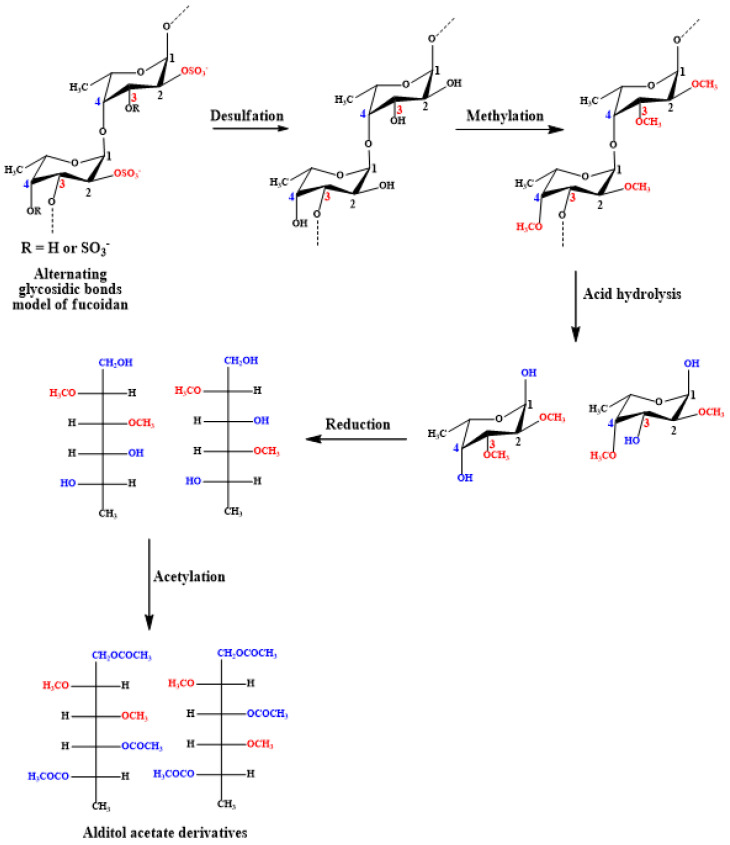
Derivatization of desulfated and deacetylated fucoidan monomers to volatile alditol acetates, which may be subsequently analyzed using GC/MS.

**Table 1 marinedrugs-18-00571-t001:** Examples of different fucoidanases and fucoidan sulfatases, including the source of catalyzed fucoidans and their mode of cleavage action.

Enzyme Source and Accession Number in GenBank *	Enzyme Substrate (Fucoidan Source)	Mode of Cleaving Actions EC Number *	Degradation Product and Structural Features	Ref.
**Fucoidanases**				
Flavobacteriacean strain, i.e., *Mariniflexilefucanivorans* SW5^T^ - FcnA: CAI47003.1 - FdlA: AAO00510.1 - FdlB: AAO00511.1	*Pelvetia canaliculata*	- *endo* *α*-(1→4) - (EC 3.2.1.212)	→3)-*α*-l-Fuc*p*, 2-OSO_3_^−^-(1→4)-*α*-l-Fuc*p*-2,3-OSO_3_^−^	[82,200,209]
*Formosa algae* (FFA1) - WP_057784217.1	*Sargassum horneri*	- *endo* *α*-(1→4) - (EC 3.2.1.212)	→3)-*α*-l-Fuc*p*, 2-OSO_3_^−^-(1→4)-*α*-l-Fuc*p*-2,3-OSO_3_^−^-(1→ fragment, with insertion of →3)-*α*-l-Fuc*p*, 2,4-OSO3^−^-(1→	[16]
	*F. evanescens*		→3)-*α*-l-Fuc*p*, 2,4-OSO_3_^−^-(1→4)-*α*-l-Fuc*p*-2,4-OSO_3_^−^-(1→4)-*α*-l-Fuc*p*, 2-OSO_3_^−^-(1→	[161]
*Formosa algae* KMM 3553^T^ (FFA2) - WP_057784219.1	*F. evanescens*	- *endo* *α*-(1→4) - (EC 3.2.1.212)	→3)-*α*-l-Fuc*p*, 2,4-OSO_3_^−^-(1→4)-*α*-l-Fuc*p*, 2-OSO_3_^−^-(1→ and →3)-*α*-l-Fuc*p*, 2-OSO_3_^−^-(1→4)-*α*-l-Fuc*p*- 2-OSO_3_^−^-(1→	[196]
			*α*-l-Fuc*p*-2-OSO_3_^−^(1→3)-*α*-l-Fuc*p*-2-OSO_3_^−^ and *α*-l-Fuc*p*-2,3-OSO_3_^−^(1→3)-*α*-l-Fuc*p*,2-OSO_3_^−^	[34]
*Fucobacter marina* SA-0082	*Kjellmaniella crassifolia* (sulfated fucoglucuronomanna)	- *endo*-*α*-d-mannosidase - (EC 3.2.1.130)	Trisaccharides composed of - Δ^4,5^Glc*p*UA-(1→2)-*α*-l-Fuc*p*, 3-OSO_3_^−^-(1→3)-*α*-d-Man*p*, - Δ^4,5^Glc*p*UA-(1→2)-*α*-l-Fuc*p*, 3- OSO_3_^−^-(1→3)-*α*-d-Man*p*, 6-OSO_3_^−^, and - Δ^4,5^Glc*p*UA-(1→2)-*α*-l-Fuc*p*, 2,4-OSO_3_^−^-(1→3)-*α*-d-Man*p*, 6-OSO_3_^−^	[210]
*Pseudoalteromonas citrea,* KMM 3296, KMM 3297, and KMM 3298 strains	*L. cichorioides* (20–40 kD)	- *endo* *α*-(1→3) - (EC 3.2.1.211)	Sulfated *α*-l-fucooligosaccharides of 1.7–5.0 kDa and 1.3–5.0 kDa by KMM 3296 and KMM 3298 strains, respectively	[211]
*Littorina kurila*	*F. distichus*	- *endo* *α*-(1→3) - (EC 3.2.1.211)	→3)-*α*-l-Fuc*p*-2,4-OSO_3_^−^-(1→4)-*α*-l-Fuc*p*-2-OSO_3_^−^-(1→	[212]
*Wenyingzhuangia fucanilytica* CZ1127^T^ - ANW96115.1 - ANW96116.1 - ANW96098.1 - ANW96097.1	*Isostichopus badionotus*	- *endo* α-(1→3) - (EC 3.2.1.211)	α-l-Fuc*p*-(1→3)-α-l-Fuc*p*(2,4-OSO_3_^–^)-(1→3)-α-l-Fucp(2-OSO_3_^–^)-(1→3)-α-l-Fuc*p*(2-OSO_3_^–^)	[202]
**Sulfatases or sulfoesterases**				
*Wenyingzhuangia fucanilytica* CZ1127^T^ - SWF1: WP_068825883.1 - SWF4: WP_068828765.1	*F. evanescens* and *S. horneri*	*exo*-2*O* and -3*O*-fucoidan sulfatase - (EC 3.1.6.B2)		[190]
*Pecten maximus*	*A. nodosum*	2*O*-fucoidan sulfatase - (EC 3.1.6.B2)		[137]

* whenever available.

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
