# Peer review of "Fucoidan Characterization: Determination of Purity and Physicochemical and Chemical Properties"

_marinedrugs, 2020, doi:10.3390/md18110571_

Round 1
Reviewer 1 Report
Additional remarks to the revised variant of the review by A.Zayed et al., Marine Drugs 997736
Lines 391-395, this text is related to monosaccharide composition of fucoidans. It should be transferred into Section 4.3.
Line 403, many fucoidans are readily soluble in DMSO, and hence, this solvent cannot be recommended for precipitation of fucoidans from solutions..
Lines 447 and 471, as it was mentioned in my previous reference, indication of purity as 95% is nonsense, since fucoidan is not a single chemical entity.
Lines 464-467, this conclusion is based on careful IR spectroscopic investigation of a series of related fucoidan molecules, and the corresponding paper should be mentioned:
M.I.Bilan, A.S.Shashkov, A.I.Usov. Structure of a sulfated xylofucan from the brown alga Punctaria plantaginea. Carbohydrate Research, 393, 1-8 (2014).
Line 479, an example of successful fractionation of a crude algal fucoidan to obtain a fucoidan fraction having regular structure, which was elucidated using NMR spectra, was described in the paper:
M.I.Bilan, A.A.Grachev, N.E.Ustuzhanina, A.S.Shashkov, N.E.Nifantiev, A.I.Usov. A highly regular fraction of a fucoidan from the brown seaweed Fucus distichus L. Carbohydrate Research, 339, 511-517 (2004).
Ref. [130] – the same paper is cited as [153].
The text contains some misprints which should be corrected.
Author Response
Kindly, refer to the attached file.

Reviewer 2 Report
The manuscript by Zayed et al. reviews the current methods for fucoidan characterization. Describing and explaining different methods used. The article scope is valuable for a broad audience interested in fucoidan research. The review shows novelty and significance.
The manuscript is however not yet ready for acceptance and needs major and extensive revising and corrections. In particular, but not only, of fucoidan chemical structures, methods and critical assessments of these, figure choice and extensive revising of section 6. Therefore I advise to reject the paper in its current form.
Hereafter is a non-exhaustive list of issues to be addressed:
Generally the methods included, should be more critically discussed. Some techniques are not frequently used, because they do not work very efficiently with fucoidans, although they work very well for other polysaccharides, like reducing end assays for instance.
Figures chosen are not consistent with the content. Why is there not more structural figures of different fucoidans, also including branchings? The structures of fucoidans should be more detailed also in the figure. Galactofucans have not been described, acetylations are lacking, branchings are lacking. Moreover, the structures of fucoidans are not completely right and the chemical fine-structures should be very thoroughly validated and corrected according to letterature. For instance according to Bilan et al. 2002, fucoidan from F. evanescens contains blocks of C2 sulfated fucoidan and blocks of C2/C4 disulfated fucoidans. C3 sulfation is rare but still present. Fig 2: A: “F. serratus L. R1=OH” is incorrect and should be “R1=H”. Fucoidans are acetylated, this has not been described in detail and has not been included in Fig. 2
Figure 1, does not give much novelty or information, furthermore fucoidan purity and co-extracted impurities are the same.
The importance of including Fig. 3 is not clear. Why is this technique singled out as important, while others are not?
Why is there a figure of alginate? This seem not relevant for the paper.
The importance of the equations included in the paper is not clear.
Seaweed species should be written out the first time, like S. polycystum page 3, line 113; L. longipes P. 4, line 141 etc.
The molecular weight of the different fucoidans have not been thoroughly explained.
Fucoidan fractionation is a very important method for purifying and separating fucoidans, this method should be better explained and not just be mentioned as part of the NMR in 5.2.
In general this paper would benefit by having a discussion of the different purification procedures in addition to the pros and cons on each with regards to purity and Mw etc. Extraction methods can change the fine-structures and Mw of fucoidans, this has not been discussed and would be good to include in this review.
HPAEC-PAD is the only technique that can distinguish between the different uronic acids and is a very important method for the investigation of fucoidan monosaccharide composition and the purity e.g. alginate contamination, but this method has not been properly described, only mentioned in one line. This method should be properly included.
Section 6 needs extensive corrections and revision. Many statements are not correct according to literature, and references needs to be double checked. Bioactivity has also been determined for some of the oligosaccharides obtained from endo-fucoidanase hydrolysis.
Some references are not correct: Line 355, should be Manns et al. 2014 (D. Manns,a A. L. Deutschle,b B. Saakec and A. S. Meyer. Methodology for quantitative determination of the carbohydrate composition of brown seaweeds (Laminariaceae). RSC Adv., 2014, 4, 25736.
Line 39: in galactofucans the ratio can be up to 1:1 fucose:galactose, so the statement that fucose predominates, is not true for all fucoidans and should be revised.
Line 120: “C2 is always sulfated” is a very conclusive statement. Since not all fucoidans have been characterized yet this perhaps is not a conclusion that can be made. Also C3 and C4 sulfation is not always present, which is not very clearly written in the text.
Line 350: “ eluted fucoidan fractions characterized by more sulfation degree and high molecular weight and vice versa” should be higher polarity, e.g. higher sulfation degree. Furthermore it is not clear what is meant by “vice versa”.
Line 176: “with the difference of those two measurements, the influence of other sugars are neglected” this sentence should be explained in more detail.
Line 282: should be rephrased and “non-real results” should be reformulated.
Line 289: both conditions are always applied”. This is incorrect. For instance, methods using hot water and microwave techniques does not necessarily use both Ca2+ precipitation and acidic treatments. Furthermore, enzymatically assisted extraction of fucoidan does not use acid.
Line 290: “alginate is always detected in crude fucoidans extracts” this is not true and a very strong statement.
Line 292:”it may interfere with uronic acid determination” this should be explained better. Alginate does not interfere with fucoidan, but the alginate will add to the amount of total uronic acids, if these are not determined by other methods like HPAEC-PAD
Line 419: Turbinaria turbinate should be corrected to Turbinaria turbinata
Line 406: why is the polarized light mentioned, when it is stated that this will give little usable information? What does the method reveal? This is not explained
Line 573: “……. makes the polymers to disulfate” what is meant by this? That the reducing environment induces di-sulfate bridge formation?
Line 602: the m/z values are clearly wrong for the suggested molecules, they should be corrected. All other m/z should be double checked
Line 608: why is the m/z different between C-2 and C-4 sulfation?
Line 631: fucoidanases does not only have activity on low weight fucoidans and what is the definition of low weight fucoidans in this respect?
Line 638: endo-fucoidanases do not cleave the 1-2 linkage in fucoidans. They cleave either 1-3 or 1-4 linkages. Although fucoidan specific fuco-glucuronomannan lyases have also been described.
Line 648: “…genes sequences are still in early stages.” What does this mean? Not many sequences of sulfatases acting on fucoidans have been found yet? Please rephrase the sentence.
Table 1: The title should be refined. Accession numbers should be given. Fda1 and 2 is lacking from the 1-3 specific fucoidanases. The recent GH168 CAZy family should be included with the characterized enzyme from Wenyingzhuangia fucanilytica CZ1127.
The viscosity of fucoidan is discussed but may very well reflect the amount of contaminating alginate left in the fucoidan extract, since the viscosity of alginate is very high, this has not been discussed.
Author Response
Kindly, refer to the attached file.

Reviewer 3 Report
The manuscript entitled "Fucoidans characterization: Determination of purity, physico-chemical, and chemical characters for reliable health-promoting benefits" is very interesting and well-designed for the aim. However it should be corrected or amended in some parts before accepted.
- The title is "Fucoidans characterization: Determination of purity, physico-chemical, and chemical characters for reliable health-promoting benefits". This article focused on mainly chemical characterization and some physico-chemical characterization. It did not revealed any relationship between structural characters and health benefits. So the term "fro reliable health-promoting benefist" could be deleted.
- In line 132-133 "R1= OSO3, R2= H; in F. serratus L.: R1= OH, R2= side chain or OSO3; Fucus evanescens C. Ag: R1= H, R2= OSO3 or H" should be corrected to "R1= SO3, R2= H; in F. serratus L.: R1= H, R2= side chain or SO3; Fucus evanescens C. Ag: R1= H, R2= SO3 or H"
- This reviewer does not understand about the figure 3. The explanation does not look sufficient. Besides a standard curve with the concentration range of fucoidan could be provided to better understand.
Author Response
Kindly, refer to the attached file.

Round 2
Reviewer 2 Report
The reviewing of the paper has indeed improved it. I although have to stress the point that the methods chosen for figure representation and thorough discussion, might pose biased self-citation. A review should be addressed in a non-biased manner and should not exclusively focus on one owns research, even if it has been published in peer-reviewed journals. To be objective and include other methods to the same extent as ones own, if those methods are used as frequent, is mandatory in order to avoid biased focus and over-citation of own work. I find this point has to be adressed and the other methods used for fucoidan characterization needs more substance.
A few specific issues needs to be adressed:
Line 362: C-PAGE can only be used to separate lower molecular weight fucoidans and not polymers, which will be retained in the top of the gel
Line 314: the crude fucoidan contained a lot of alginate using the enzyme-assisted method, so the statement here is incorrect
Line 678: it seems like something is missing in the line, what is mean with “aside from”? aside from what?
Line 685: previous sentences refer to both fucoidanases and sulfatases, although this line only apply to fucoidanases, since the sulfatases are not preserving the sulfation pattern
Line 698: EC 3.2.1.44 has in EXPASY been transferred to EC 3.2.1.211, please check and rephrase accordingly. The specificity of these enzymes are not alpha 1-2 linkages but 1-3. Furthermore, the fucoidanases acting endo also includes EC 3.2.1.212, e.g. GH107
Line 705: fucoglucoronnomannan lyases were first described to be cleaving linkages between mannose and glucuronic acids in a lyase manner, by Takayama et al. 2002 Patent number: US 6,489,155 B1 and Sakai et al 2003. They were later analyzed on different fucoidan substrates in Cao et al. 2018 and suggested to cleave alpha 1-4 linkages between fucose residues, although this was not investigated by other methods than C-PAGE.
Line 716-18: Bioactivity was not investigated in Cao et al. 2018. The FcnA was produced in a more stable form, after increased C-terminal deletion in Cao et al 2018. And the enzyme was shown to act on several different fucoidans.
Table 1: Some accession numbers are still missing: Fucoglucuronomannalyases FdlA: AAO00510.1 and FdlB: AAO00511.1. Accession numbers of the sulfatases from Wenyingzhuangiafucanilytica CZ1127T should also be added.
Author Response
Dear Reviewer#2,
The authors would like to thank you for your diligent effort in improving the quality of our manuscript. Kindly, find enclosed a detailed rebuttal list discussing the raised points.
Kind regards,
Authors of the manuscript

Round 3
Reviewer 2 Report
Thank you for your reply. I have one minor correction:
Line 696: EC 3.2.1.211 is believed to cleave 1-3, while EC 3.2.1.212 likely cleave 1-4 linkages, please correct
Author Response
Done and highlighted. The information was also mentioned in Table 1.
Thank you and kind regards,
On behalf of the manuscript's authors
This manuscript is a resubmission of an earlier submission. The following is a list of the peer review reports and author responses from that submission.
Round 1
Reviewer 1 Report
The review is devoted to description of methods of characterization of fucoidans. Despite the fact that similar reviews appeared recently in many journals, it may be considered as a potential source of references to new publications. At the same time, it should be noted that the text deserves serious critical remarks.
Page 1, keywords. Evidently physico-chemical properties, structural elucidation and especially bioactivities need more detailed description.
Page 1, section 1. The term ‘fucoidans’ should be retained only for complex polysaccharides of algal origin, whereas polysaccharides built up of fucose and sulfate only, obtained from marine invertebrates, may be designated, according to the Carbohydrate Nomenclature, as sulfated fucans or fucan sulfates, FS. Investigation of these two groups of fucose-containing biopolymers requires different approaches. It is reasonable to describe them separately in order to emphasize the substantial differences between structural analysis of regular linear homopolysaccharides and non-regular, mostly branched heteropolysaccharides.
Page 3-4, section 2, Fig. 2. Recent studies demonstrated more variable backbone structures, including chains of B-type in Fucales etc. Structural differences may be found even between fucoidans isolated from very closely related species of brown seaweeds.
Page 4-5, section 3.1.1. It is not necessary to describe in detail the most popular reaction of sugars with phenol and sulfuric acid and to illustrate it by speculative Fig. 3.
Page 5, section 3.1.2 is omitted.
Page 5, section 3.1.3. In addition to detailed description of analytical procedure, it is important to mention that algal polyphenols may interfere to a great extent in colorimetric fucose determination.
Page 5, section 3.1.4. It should be certainly mentioned that the usual problem in quantitative determination of fucoidan content is the absence of appropriate standard. Commercial preparations may be insufficiently purified and may be structurally different from analytical samples.
Page 6, line 200. Commercial fucoidans are 95% pure – what does it mean? Since fucoidans are not individual chemical substances, this percentage is a nonsense.
Page 6, section 3.1.4 – previous section is also numbered as 3.1.4.
Page 6, line 209. “Sulfate ester groups are highly susceptible to hydrolysis” – this is incorrect, they are even more stable than most fucosidic linkages. Nevertheless, turbidimetric analysis requires preliminary liberation of sulfate by acid hydrolysis.
Page 7, section 3.1.5. Identification of uronic acids is necessary to distinguish components of fucoidan from components of alginic acids.
Page 9, line 334, section 4.2 – previous section is also numbered as 4.2.
Pages 9-10 and Fig. 5 – methylation is described as monomeric composition analysis, but this method is used to determine positions of inter-residue linkages and sulfate groups. Desulfation procedures are mentioned without discussion.
Page 11, section 5.1. Structural information containing in FT-IR spectra is not very valuable. It should be noted that positions of signals of secondary sulfate groups depend on real conformation of monosaccharide units, which may be considerably distorted in branched and heavily sulfated chains by neighboring substituents.
Page 12, section 5.3. NMR spectra can be interpreted only for regular (or masked regular) polymeric structures. In the case of algal fucoidans such structures may be obtained by careful fractionation of crude fucoidan or by specific chemical modification, such as desulfation. Both approaches are not described and discussed in the text.
Page 13, Fig. 7. This picture contains the formula of 3-linked D-fucose (it is terrible!) and a spectrum of commercial fucoidan (is it 95% pure?) containing absolutely unresolved anomeric region. Proton signals of fucose methyl groups are also unresolved. How can you use such spectrum in structural analysis?
Page 14, section 5.5. Mass spectrometry is used for analysis of oligosaccharide fragments of fucoidans. Hence, it is desirable to mention here the methods of partial depolymerization of fucoidans and to give examples of interpretation of mass spectra.
Conclusion. The review contains a lot of drawbacks and cannot be accepted for publication in the present form. It should be returned to the authors for major improvement or even rejected.
Reviewer 2 Report
The review entitled: “Fucoidans characterization: Purity, physico-chemical and chemical properties” describes how to characterize a fucoidan polysaccharide. The description of the methods usually used to reach this goal are well reported, while few physico-chemical properties are analyzed. Although the review is well written and set up, it lacks novelty and originality. I think it should be rejected.
Reviewer 3 Report
Fucoidans characterization: Purity, physico-chemical and chemical properties by Zayed, A. et. al.
Due to highly variable chemistry of fucoidans, they have diverse functionalities. Authors have used various techniques (e.g., FTIR, NMR, MS and HPLC) to elucidate the structural features, such as position of sulfate ester groups and glycosidic bonds. This study will help understanding the fucoidan mechanism of action and exact interaction with different human targets.
This is detailed study, well written and easily understandable.